# Risk factors of suicidality among married adults: A cross-sectional survey in Rajshahi City, Bangladesh

Abdul Wadood[1]☉, Rezaul Karim[2‡], Abdullah Al Mamun Hussain[3‡], Masud Rana[4‡], Golam Hossain[5]☉ *

1 Medical Centre, University of Rajshahi, Rajshahi, Bangladesh, 2 Department of Biochemistry and Molecular Biology, University of Rajshahi, Rajshahi, Bangladesh, 3 Department of Psychiatry, Rajshahi Medical College, Rajshahi, Bangladesh, 4 Department of Population Science and Human Resource Development, University of Rajshahi, Rajshahi, Bangladesh, 5 Department of Statistics, Health Research Group, University of Rajshahi, Rajshahi, Bangladesh

☉ These authors contributed equally to this work.
‡ These authors also contributed equally to this work.
* hossain95@yahoo.com

**Data Availability Statement:** All relevant data are within the paper and its Supporting information files.

## Abstract

### Background

Suicide is a serious public health concern all over the world including Bangladesh. About 9% of the patients admitted with suicidal ideation (SI) or suicide attempt (SA) later complete suicide. To understand and prevent suicide, the study of SI and SA is necessary but research in this area is scanty in Bangladesh. Therefore, we studied suicidality (SI and SA) among married adults in Rajshahi City, Bangladesh.

### Methods

This was a household cross-sectional study. A total of 708 married adults were selected for this study using a multi-stage random sampling. Suicidality was measured based on two factors: (i) suicidal ideation, and (ii) suicide attempt. Frequency distribution, Chi-square test and multiple binary logistic regression model were used in this study according to our objectives.

### Results

The prevalence of suicidal ideation, suicide attempt, and suicidality was 5.8%, 3.4%, and 8.3% respectively among married adults. A multiple binary logistic regression model provided the following risk factors of suicidality: (i) joint family (AOR = 0.310, p<0.01), (ii) ≥26 years of age at the first marriage (AOR = 0.379, p<0.05), (iii) twice or more marriage (AOR = 0.214, p<0.01), (iv) conjugal life of ≥16 years (AOR = 0.410, p<0.05), (v) having no child (AOR = 6.343, p<0.01) and (vi) having 1–2 children (AOR = 6.190, p<0.01), (vii) medical comorbidity (AOR = 0.421, p<0.01), (viii) mental comorbidity (AOR = 0.253, p<0.01), (ix) stress-anxiety (AOR = 0.311, p<0.01), (x) family history of mental disorders (AOR = 0.059, p<0.01), (xi) family history of suicide/suicide attempt (AOR = 0.009, p<0.01), (xii) substance

**Funding:** The authors received no specific funding for this work.

**Competing interests:** The authors have declared that no competing interests exist.

abuse (AOR = 0.065, p<0.01), (xiii) poor relationship with spouse (AOR = 0.209, p<0.01), and (xiv) poor relationship with other family members (AOR = 0.347, p<0.05).

## Conclusion

The prevalence of suicidality is remarkable in Rajshahi city, Bangladesh. The government and non-government agencies can use the findings of this study to identify the vulnerable groups and undertake measures for preventing and reducing suicidality.

## Introduction

Suicide is a serious public health concern all over the world. Global yearly deaths due to suicide estimate at about 800,000 people [1]. To understand and prevent suicide, the study of suicidal ideation (SI) and suicide attempt (SA) is necessary, as suicide is highly predicted by suicide attempt (SA) [2,3] and 8.6% of the patients admitted with suicidal ideation (SI) or SA would later complete suicide [4]. In the USA, 80% of attempters completed suicide within the next one year [5]. There are estimates of more than 20 SAs preceding one completed suicide [6]. SA was found to be the single most predictor of completed suicide in many other studies also [7–11]. The actual and complete picture of SAs cannot be known because most of the incidents are kept unreported or hidden by the families to avoid social and legal consequences and the cases coming to the healthcare service centers are only the tip of the iceberg [6,12–14]. On the other hand, SI is an important predictor of SA. SI was found to be highly correlated to SA in the future [15–17]. An Indian study found 24.6% and 7.1% of the adult population having SI and SA [18]. According to a population-based study in rural central India, the past 6-month prevalence of SI and SA was 5.1% and 4.2% respectively [19]. Among Pakistani medical students, 31.4% had SI [20]. A later study reported that 35.6% of Pakistani medical students had SI, 13.9% planned to commit suicide, and 4.8% attempted suicide [21]. In Sri Lanka, the risk of SA significantly varied between 21.0% in households and 4.0% in communities [22]. Another study found that 35.3% of the Nepalese population had high intent of committing suicide and 6.0% attempted suicide [23]. The life-time prevalence of SI was 20.0% among medical students in Nepal [24] and 3.1% among the general population in Bhutan [25].

A study conducted in the Indian subcontinent countries revealed that 10% of the number of suicides in the world was committed in India, Pakistan and Sri Lanka and suicides in other countries of the subcontinent (Bangladesh, Nepal, Afghanistan, Bhutan, and the Maldives) were not well-documented [26]. A review study demonstrated that in six South Asian countries (Afghanistan, Bangladesh, India, Nepal, Pakistan, Sri Lanka) the reported suicide rates ranged from 0.43 (Pakistan) to 331.0 (India) per 100,000 population, the non-pooled average rate of suicide being 25.2 (58.3 in Bangladesh) [27]. An epidemiological study found that the overall rate of suicide was 7.3 per 100,000 per year in Bangladesh [28].

A study estimated that 28.18% of people attempted suicide in the past year of the survey in Bangladesh [29]. On the other hand, 23.0% of elderly people in rural area of Bangladesh had suicidal ideation [30]. Another study observed that the prevalence of suicidal ideation was 5.0% among adolescents (14–19 years) in Bangladesh [31]. According to another study, 11.0–14.0% of ever-married women of reproductive age in Bangladesh reported suicidal ideation [32].

The main risk factors for suicide in the USA are summarized as: prior suicide attempt, mental disorders, substance abuse disorder, family history of a mental health, family history of substance abuse disorder, family history of suicide, family violence, having guns or other firearms

in the home, being in prison or jail, being exposed to others' suicidal behavior, medical illness, being between the ages of 15 and 24 years or over age 60, social isolation, criminal problems, financial problems, impulsive or aggressive tendencies, job problems or loss, relationship problems such as a break-up, violence, or loss, and sexual violence [33–36]. Though most of these factors are also common, the high use of organophosphate pesticides, larger numbers of married women, fewer elderly people, and problems of interpersonal relationship and adverse life events appear as additional important risk factors for suicide in the Indian subcontinent [26]. A review study found that emotional stress due to quarrel in family, poverty, long-term illness, failure in exam, suicidal death in near relative, substance abuse, stressful events, history of criminal behavior, social deprivation, uncertainty about future, not raised by biological parents and marital disharmony are the most common risk factors of suicide in Bangladesh [37].

However, to the best of our knowledge, the study on suicidality (SI and SA) among urban married adults in Bangladesh is not available. Usually, in Bangladesh, married adults are the key persons for earning livelihood and maintain their families. Suicidality among them disrupts the whole family. The situation is worse in urban area because married adults depend mainly on business and services for their livelihood. Due to their importance in family, it is necessary to investigate the suicidality among them.

Therefore, the objectives of this study were: (i) to determine the prevalence of suicidal ideation, suicide attempt and the overall suicidality, and (ii) to investigate the associated factors of suicidality among married adults in Rajshahi City, Bangladesh.

## Materials and methods

### Study design, setting, and population

This was a household-based cross-sectional study and Rajshahi City area was our study area. Rajshahi City is one of the four biggest cities in Bangladesh and the headquarters of Rajshahi Division, the north-western region of the country. The city consists of 97.17 square kilometers area and is divided into 30 Wards (administrative units). It comprises 99,222 households [29]. Married adults living in this city were considered as our study population. Only legally married adults aged at least 18 years were included in the study. The survey was conducted from May 15 to July 30, 2019. The present study was a part of our project.

### Sample size determination and sampling

For determining the sample size, the mathematical formula $n = N/(1+Nd^2)$ was used in this study, where n = sample size, N = population size, and d (margin of error) = 0.05. The formula showed that 398 households would suffice for this study. We assumed a 90% rate of response and decided to select 450 households. For sample selection, a multistage random sampling technique was applied. Before sampling, we went to the Rajshahi City Corporation Office and collected the necessary information about Wards, muhallahs, and households. Then, at first, we selected three Wards by a simple random sampling (lottery), in the second stage, three muhallas from each selected Ward by lottery, and lastly, 50 households from each selected muhalla again by lottery. One married male and one married female were selected from each selected household. We selected them by lottery when there were more than one married male and one married female in a household.

### Ethical approval

The ethical approval for our study was given by the Institutional Animal, Medical Ethics, Biosafety and Biosecurity Committee (IAMEBBC) for Experimentation on Animal, Human,

Microbes and Living Natural Sources, Institute of Biological Sciences, University of Rajshahi, Bangladesh (Memo No: 120/ 320/ IAMEBBC/ IBSc, dated April 11, 2019).

## Data collection

A self-developed semi-structured questionnaire was used for data collection in this study. The questionnaire included demographic, socioeconomic, and comorbidity-related questions and statements. The original questionnaire was drafted in English and then translated to Bangla, the mother language of Bangladesh to make it easily comprehensible. The first author prepared the first drafts of both English and Bangla versions of the questionnaire and the other authors reviewed, edited, and improved it. A total of six interviewers were trained and employed for data collection. Under the direct supervision of the first and last authors, the interviewers went to the selected households and briefed the selected married adults about the study aims and procedures. We had a target of collecting information from 900 respondents from the selected 450 households. But 73 married adults disagreed to give interviews, and 66 others were not available in their households at the time of the survey. Thus, a total of 139 (73+66 = 139) respondents were excluded from the study. Written informed consent was taken from the rest 761 (900–139 = 761) agreed respondents and they were interviewed face-to-face. While entering data into the computer, 53 questionnaires were discarded for missing information. Finally, we had data of 708 respondents for analysis.

## Variables

In the current study, suicidality was the principal outcome variable and it was measured based on two factors: (i) suicidal ideation, and (ii) suicide attempt. Suicidal ideation was assessed by the response of the respondents to a question 'Did you ever think of committing suicide in the last six months?' and suicide attempts by the response to a question 'Did you ever attempt to commit suicide in the last six months?' Each of these two questions was assigned two options of the answer– 'no' (score 0) and 'yes' (score 1). Suicidality was measured by addition of suicidal ideation and suicide attempt, the sum of score would be 0, or 1, or 2 of a respondent. We classified our sample into two groups; (i) no suicidality (sum of score 0; code 0), and (ii) suicidality (sum of score 1–2; code 1). We also considered suicide attempt as an outcome variable and suicidal ideation as an independent variable to examine the association between these two factors and assess the impact of suicidal ideation on suicide attempt among married adults. For identifying risk factors of suicidality, we considered some well-used household, demographic, socioeconomic, familial, and comorbidity-related factors as independent variables in this study (Table 1).

## Statistical analysis

The data were presented as mean and standard deviation (SD) for the numerical variables and frequency with percentage (%) for the categorical variables. We applied descriptive analysis for determining the prevalence of suicidal ideation, suicide attempt, suicidality, and the sample characteristics, the chi-square test for examining the associated factors of suicidality, and multivariable binary logistic regression model for identifying the most contributory risk factors of suicidality and impact of suicidal ideation on suicide attempt. For data analysis, SPSS (IBM, version 22) was used. Statistical significance was accepted at $p < 0.05$ and a 95% confidence interval (CI) were set for the odds ratio.

**Table 1. Selected independent variables with their categories, definition, codes and frequency distribution.**

| Variable | Category | N (%) |
|---|---|---|
| Age (in year) | 18–25 | 188 (26.55) |
| | 26–35 | 281 (39.69) |
| | ≥36 | 239 (33.76) |
| Gender | Male (husband) | 354 (50.0) |
| | Female (wife) | 354 (50.0) |
| Education level | Uneducated | 75 (10.59) |
| | Primary | 216 (30.51) |
| | Secondary | 204 (28.81) |
| | Higher | 213 (30.09) |
| Respondent's occupation | Homemaker (housewife) | 316, (44.63) |
| | Hard-worker (labor and farmer) | 211 (29.81) |
| | Service | 181 (25.56) |
| Respondent's nutritional status | Under-nourished (BMI<18.5 kg/m$^2$) | 40 (5.65) |
| | Healthy (18≤BMI<25 kg/m$^2$) | 443 (62.57) |
| | Over-nourished (BMI≥25 kg/m$^2$) | 225 (31.78) |
| Monthly family income | Poor (≤20,000 Taka) | 225 (31.78) |
| | Middle (20,001–40,000 Taka) | 419 (59.18) |
| | Rich (≥40,001 Taka) | 30 (4.24) |
| Type of family | Nuclear (family of parents and children) | 524 (74.01) |
| | Joint (family of parents, children, grandparents, grandchildren and relatives) | 194 (27.40) |
| Number of ever-born children | No child | 86 (12.15) |
| | 1–2 children | 443 (62.57) |
| | ≥3 children | 179 (25.28) |
| Duration of present conjugal life (year) | ≤5 | 214 (30.23) |
| | 6–15 | 306 (43.22) |
| | ≥16 | 188 (26.55) |
| Number of marriage | One | 687 (97.03) |
| | 2 or more | 21 (2.97) |

*(Continued)*

**Table 1.** (Continued)

| Variable | Category | N (%) |
|---|---|---|
| Age at the first marriage (year) | <18 | 198 (27.97) |
| | 18–25 | 435 (61.44) |
| | ≥26 | 75 (10.59) |
| Death of children | No | 628 (88.70) |
| | Yes | 80 (11.30) |
| Stress-anxiety | No | 520 (73.45) |
| | Yes | 188 (26.55) |
| Your relationship with the spouse | Poor | 23 (3.25) |
| | Good | 685 (96.75) |
| Your relationship with other family members | Poor | 34 (4.80) |
| | Good | 674 (95.20) |
| Medical comorbidity | No | 614 (86.72) |
| | Yes | 94 (13.28) |
| Mental comorbidity | No | 677 (95.62) |
| | Yes | 31 (4.38) |
| Family history of mental disorders | No | 684 (95.62) |
| | Yes | 24 (4.38) |
| Substance abuse | No | 662 (93.50) |
| | Yes | 46 (6.50) |
| Family history of suicide/suicidal attempt | No | 667 (94.21) |
| | Yes | 41 (5.79) |

## Results

A total number of 708 married adults living in Rajshahi city, Bangladesh were considered as sample to investigate suicidality in last six-month of our survey. The mean age of our subject was 33.21±10.39 years with age range from 18 to 82 years. 10.6% and 30.1% married adults were uneducated and higher educated respectively. The prevalence of medical and mental comorbidities among adults was 32.1% and 4.4% respectively. It was observed that 44.6%, 29.8% and 25.6% married adults were homemakers, hard workers and service holders respectively (Table 1). The mean duration of their conjugal life was 11.23±8.81 years with range from 1 to 45 years.

### Prevalence of suicidal ideation, suicide attempt, and suicidality

The prevalence rate of suicidality was found to be 8.3%, 9.0%, and 7.6% among married adults, males, and females respectively (Fig 1). The same figure shows that the prevalence of suicidal

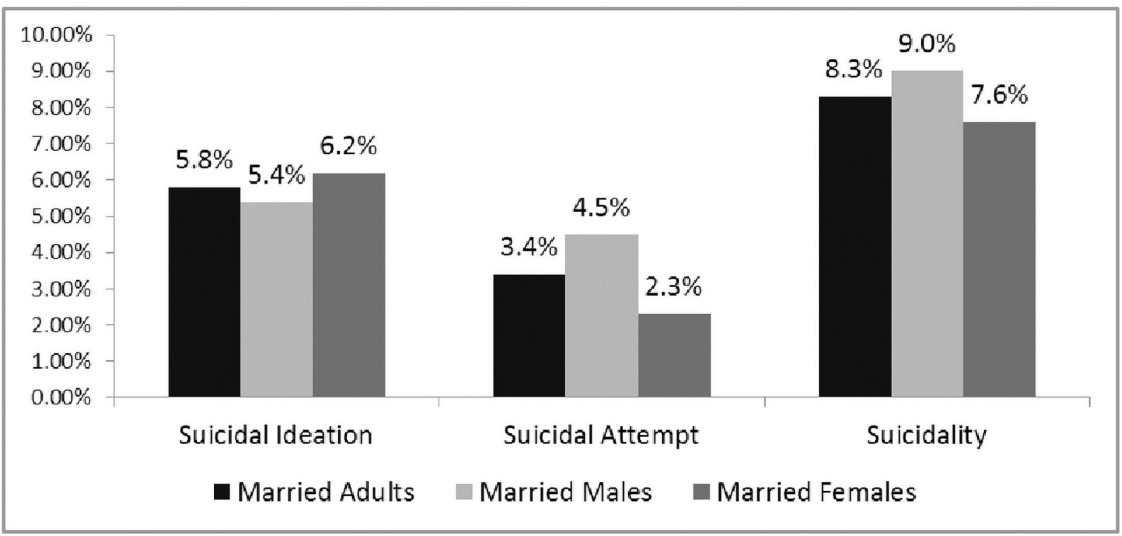

**Fig 1. Prevalence of suicidal ideation, suicidal attempt, and suicidality among married adults in Rajshahi City, Bangladesh.**

ideation was 5.8%, 5.4%, and 6.2%, and of suicide attempt was 3.4%, 4.5%, and 2.3% among married adults, males, and females respectively (Fig 1).

## Association of suicidality

The chi-square test revealed that type of family, number of ever-born children, age at the first marriage, number of marriage, duration of conjugal life, death of children, medical comorbidity, mental comorbidity, family history of mental disorders, stress-anxiety, family history of suicide/attempt, substance abuse, relation with the spouse, and relation with other family members were significantly ($p < 0.05$) associated with suicidality among married adults in Rajshahi City, Bangladesh (Table 2).

## Effect of associated factors on suicidality

The statistically significant associated factors (14 in number) found in the chi-square test (p-value $< 0.05$) were put as independent variables in the multivariable binary logistic regression analysis. Being member of the joint family ($p < 0.01$), $\geq 26$ years of age at the first marriage ($p < 0.05$), twice or more marriage ($p < 0.01$), conjugal life of $\geq 16$ years ($p < 0.05$), having no child ($p < 0.01$) and 1–2 children ($p < 0.01$), medical comorbidity ($p < 0.01$), mental comorbidity ($p < 0.01$), stress-anxiety ($p < 0.01$), family history of mental disorders ($p < 0.01$), family history of suicide/attempt ($p < 0.01$), substance abuse ($p < 0.01$), poor relationship with spouse ($p < 0.01$), and poor relationship with other family members ($p < 0.05$) were found to show high odds of likelihood of developing suicidality among married adults (Table 3).

## Impact of suicidal ideation on suicide attempt

After controlling the effect of other selected factors, multiple logistic regression model showed that married adults having suicidal ideation had 83.8% of higher chance to attempt suicide than their counterparts (AOR = 0.162, 95% CI = 0.060–0.433, $p < 0.01$) (Table 4).

**Table 2. Association between selected independent variables and suicidality among married adults (n = 708).**

| Variables, Category | Suicidality N (%) | No Suicidality N (%) | $\chi^2$-value | p-value |
|---|---|---|---|---|
| **Gender** | | | 0.462 | 0.587 |
| Male | 32 (9.0) | 322 (91.6) | | |
| Female | 27 (7.6) | 327 (92.4) | | |
| **Age (Year)** | | | 0.749 | 0.701 |
| 18–25 | 13 (6.9) | 175 (93.1) | | |
| 26–35 | 24 (8.5) | 257 (91.5) | | |
| ≥36 | 22 (9.2) | 217 (90.8) | | |
| **Education level** | | | 5.400 | 0.144 |
| Uneducated | 9 (12.0) | 66 (88.0) | | |
| Primary | 11 (5.1) | 205 (94.9) | | |
| Secondary | 17 (8.3) | 187 (91.7) | | |
| Higher | 22 (10.3) | 191 (89.7) | | |
| **Respondent's occupation** | | | 0.521 | 0.777 |
| Homemaker | 25 (7.9) | 291 (92.1) | | |
| Hard worker | 20 (9.5) | 191 (90.5) | | |
| Service | 14 (7.7) | 167 (92.3) | | |
| **Monthly family income (Taka)** | | | 2.735 | 0.284 |
| Poor, ≤20,000 | 48 (9.3) | 466 (90.7) | | |
| Middle, 20,001–40,000 | 10 (6.1) | 154 (93.9) | | |
| Rich, ≥40,001 | 1 (3.3) | 29 (96.7) | | |
| **Respondent's nutritional status** | | | 2.354 | 0.321 |
| Under-nourished | 3 (7.5) | 37 (92.5) | | |
| Healthy | 32 (7.2) | 411 (92.8) | | |
| Over-nourished | 24 (10.7) | 207 (89.3) | | |
| **Type of family** | | | 17.787 | 0.001 |
| Nuclear | 29 (5.6) | 485 (94.4) | | |
| Joint | 30 (15.5) | 165 (84.5) | | |
| **Number of ever-born children** | | | 11.512 | 0.003 |
| No child | 12 (14.0) | 74 (86.0) | | |
| 1–2 children | 42 (9.5) | 401 (90.5) | | |
| ≥3 children | 5 (2.8) | 174 (97.2) | | |
| **Age at the first marriage (Year)** | | | 7.885 | 0.019 |
| <18 | 9 (4.5) | 189 (95.5) | | |
| 18–25 | 39 (9.0) | 396 (91.0) | | |
| ≥26 | 11 (14.7) | 64 (85.3) | | |
| **Number of marriage** | | | 6.786 | 0.024 |
| One | 54 (7.9) | 633 (92.1) | | |
| 2 or more | 5 (23.8) | 16 (76.2) | | |
| **Duration of present conjugal life (year)** | | | 9.981 | 0.007 |
| ≤5 | 27 (12.6) | 187 (87.4) | | |
| 6–15 | 15 (4.9) | 291 (95.1) | | |
| ≥16 | 17 (9.0) | 171 (91.0) | | |
| **Death of child** | | | 5.247 | 0.025 |
| No | 47 (7.5) | 581 (92.5) | | |
| Yes | 12 (15.0) | 68 (85.0) | | |
| **Medical comorbidity** | | | 16.598 | 0.001 |
| No | 41 (6.7) | 573 (93.3) | | |

*(Continued)*

**Table 2.** (Continued)

| Variables, Category | Suicidality N (%) | No Suicidality N (%) | $\chi^2$-value | p-value |
|---|---|---|---|---|
| Yes | 18 (19.1) | 76 (80.9) | | |
| **Mental comorbidity** | | | 27.408 | 0.001 |
| No | 25 (4.9) | 483 (95.1) | | |
| Yes | 34 (17.0) | 166 (83.0) | | |
| **Family history of mental disorders** | | | | |
| No | 47 (6.9) | 637 (93.1) | 56.459 | 0.001 |
| Yes | 12 (50.0) | 12 (50.0) | | |
| **Stress-Anxiety** | | | 19.478 | 0.001 |
| No | 29 (5.6) | 491 (94.4) | | |
| Yes | 30 (16.0) | 158 (84.0) | | |
| **Family history of suicide/attempt** | | | 289.938 | 0.001 |
| No | 22 (3.4) | 634 (96.6) | | |
| Yes | 37 (71.2) | 15 (28.8) | | |
| **Substance abuse** | | | 100.447 | 0.001 |
| No | 37 (5.6) | 625 (94.4) | | |
| Yes | 22 (47.8) | 24 (52.2) | | |
| **Relationship with spouse** | | | 29.516 | 0.001 |
| Good | 50 (7.3) | 635 (92.7) | | |
| Poor | 9 (39.1) | 14 (60.9) | | |
| **Relationship with other family members** | | | 10.773 | 0.001 |
| Good | 49 (7.3) | 625 (92.7) | | |
| Poor | 10 (29.4) | 24 (70.6) | | |

## Discussion

In this study, we determined the six-month prevalence of suicidal ideation, suicide attempt, and overall suicidality (both SI and SA), identified the risk factors of suicidality, and the effect of suicidal ideation on suicide attempt among married adults living in Rajshahi City, Bangladesh.

### Prevalence of suicidal ideation, suicide attempt, and suicidality

The findings of this study regarding the six-month prevalence of SI and SA among married adults were remarkably less than that found in a few previous studies with other types of populations in Bangladesh [29–31]. However, in the context of other countries, the findings of this study were acceptable. In the USA, incidents of SA were 0.79% in 2013 [38]. Another USA study reported 3.8% of adults had SI in the last one year [16]. The 12-month prevalence of SI was 3.4% in Australia [17]. An Indian study found 24.6% of SI and 7.1% of SA among the adult population [18]. According to another Indian study, the past 6-month prevalence of SI and SA was 5.1% and 4.2% respectively [19]. The prevalence of SI and SA was much higher among Pakistani medical students [20–21] and in Sri Lanka, the risk of SA significantly varied between 21.0% in households and 4.0% in communities [22]. Nepalese studies found that the rate of SI was higher in that country, 35.3% among the general population [23], and 20.0% among medical students [24]. However, in Bhutan, SI was found in 3.1% of the general population [25].

In a rural area in Bangladesh, 28.18% of people attempted suicide in the past year [29]. In another study among elderly people in rural Bangladesh, suicidal ideation was found to be

**Table 3. Effect of the associated factors on suicidality among married adults (n = 708) in Rajshahi City, Bangladesh.**

| Variables | B | S. E | p-value | AOR | 95% CI for AOR | |
|---|---|---|---|---|---|---|
| | | | | | Lower | Upper |
| **Type of family** | | | | | | |
| Nuclear versus Joint[R] | -1.172 | 0.305 | 0.001 | 0.310 | 0.170 | 0.563 |
| **Age at the first marriage** | | | | | | |
| <18 years versus ≥26 years[R] | -0.970 | 0.493 | 0.049 | 0.379 | 0.144 | 0.998 |
| 18–25 years ≥26 years[R] | -0.199 | 0.388 | 0.608 | 0.820 | 0.383 | 1.754 |
| **Number of marriage** | | | | | | |
| One versus ≥2[R] | -1.540 | 0.591 | 0.009 | 0.214 | 0.067 | 0.682 |
| **Duration of conjugal life** | | | | | | |
| ≤5 years ≥16 years[R] | -0.204 | 0.408 | 0.617 | 0.816 | 0.367 | 1.814 |
| 6–15 years ≥16 years[R] | -0.890 | 0.399 | 0.026 | 0.410 | 0.188 | 0.898 |
| **Death of child** | | | | | | |
| No versus Yes[R] | -0.481 | 0.383 | 0.209 | 0.618 | 0.292 | 1.308 |
| **Number of ever-born children** | | | | | | |
| No child versus ≤3 Children[R] | 1.847 | 0.649 | 0.004 | 6.343 | 1.777 | 22.643 |
| 1–2 children versus ≤3 Children[R] | 1.823 | 0.538 | 0.001 | 6.190 | 2.156 | 17.772 |
| **Medical comorbidity** | | | | | | |
| No versus Yes[R] | -0.864 | 0.274 | 0.002 | 0.421 | 0.246 | 0.721 |
| **Mental comorbidity** | | | | | | |
| No versus Yes[R] | -1.376 | 0.278 | 0.001 | 0.253 | 0.146 | 0.436 |
| **Stress-Anxiety** | | | | | | |
| No versus Yes[R] | -1.168 | 0.276 | 0.001 | 0.311 | 0.181 | 0.534 |
| **Family history of mental disorder** | | | | | | |
| No versus Yes[R] | -2.835 | 0.594 | 0.001 | 0.059 | 0.018 | 0.188 |
| **Family history of suicide/attempt** | | | | | | |
| No versus Yes[R] | -4.766 | 0.462 | 0.001 | 0.009 | 0.003 | 0.021 |
| **Substance abuse** | | | | | | |
| No versus Yes[R] | -2.740 | 0.340 | 0.001 | 0.065 | 0.033 | 0.126 |
| **Relationship with spouse** | | | | | | |
| Good versus Poor[R] | -1.567 | 0.528 | 0.003 | 0.209 | 0.074 | 0.587 |
| **Relationship with family members** | | | | | | |
| Good versus Poor[R] | -1.059 | 0.486 | 0.029 | 0.347 | 0.134 | 0.899 |

N. B.: B-Coefficient; S. E. -Standard Error; AOR -Adjusted Odds Ratio; CI -Confidence Interval;

[R] -Reference.

**Table 4. Impact of suicidal ideation on suicide attempt among married adults in Rajshahi City, Bangladesh.**

| Variables | B | S. E | p-value | AOR | 95% CI for AOR | |
|---|---|---|---|---|---|---|
| | | | | | Lower | Upper |
| **Suicidal Ideation** | | | | | | |
| No versus Yes[R] | -1.821 | 0.502 | 0.001 | 0.162 | 0.060 | 0.433 |

N. B.: B-Coefficient; S. E. -Standard Error; AOR -Adjusted Odds Ratio; CI -Confidence Interval;

[R] -Reference.

23.0% [30]. Another study observed that the prevalence of suicidal ideation was 5.0% among adolescents of 14–19 years in Bangladesh [31]. Compared to these Bangladeshi findings, the prevalence of SI and SA in our study was remarkably less. The urban setting, the prevalence of last six months, and married status of the respondents in our study might be the reason for such a big difference. In a Muslim majority country like Bangladesh, suicide attempt and suicide are considered sin, and the person, and even the family as a whole, is socially stigmatized. Suicide and suicide attempt are also a crime in this country. That is why people try to keep most of the incidents unreported or hidden for avoiding social stigma and legal consequences [6, 12–14]. That might be a reason for the lower rate of SI and SA in our study. The prevalence of suicidality as a whole cannot be compared as it was not studied yet in Bangladesh, so far our knowledge goes.

## Risk factors of suicidality

In this study, we found that joint family, ≥26 years of age at the first marriage, twice or more marriage, conjugal life of ≥16 years, having no child and ≥3 children, medical comorbidity, mental comorbidity, stress-anxiety, family history of mental disorders, family history of suicide/attempt, substance abuse, poor relationship with the spouse, and poor relationship with other family members contributed more to develop suicidality among married adults in Bangladesh. These factors fall into different categories such as individual, social, community, interpersonal relationships within the family, and also the healthcare system, and overlap one another [39]. Mentionable that risk factors differ in their importance in different countries and regions, and even within a country or region, differences are found in the prevalence of risk factors [40]. Some factors such as the increasing use of the internet and migration of people from one place to another place in the same country or region also influence the prevalence of risk factors [41]. Some risks were universal but some other risk factors in low-income countries differ from high-income countries [42–43]. In African countries, a study reported, difficult interpersonal relationships, physical and mental comorbidities, socioeconomic problems, and substance abuse were the common risk factors of suicide and SA [44] while in Latin America and the Caribbean, major depression, family dysfunction, and previous SA were found to be the common risk factors of SA [45]. Another study included younger and older age, mental comorbidity, low socioeconomic status, substance abuse, and prior SA as the universal risk factors of suicide [40]. Stress was also a risk factor common in both low- and high-income countries [43]. Depressive disorder, mental comorbidities, substance abuse or dependence were potential risk factors of suicide in low-income countries [46]. A Nigerian study reported that physical comorbidity was correlated to suicide [47]. History of suicide in the family is highly associated with suicidality [48,49]. An Indian study reported that difficult relationship with spouse and family members, substance abuse, and mental comorbidity significantly contributed to committing suicide [50]. Family conflicts and mental comorbidities were found to contribute more to suicide attempts in a study [51]. A recently published study observed that physical and mental comorbidities were important predictors of suicide attempts [52]. 12-month suicidal ideation and suicide attempts were associated with mental disorders like anxiety and depression and substance abuse [53]. Findings of all the above-mentioned studies reveal that physical and mental comorbidities, substance abuse, stress-anxiety, family history of suicide/attempt, and family conflicts are the universal predictors of suicidality that corroborates our findings. However, there are contradictions regarding other findings. Joint family, ≥26 years of age at the first marriage, twice or more marriage, conjugal life of ≥16 years, and having no child overlap one another and ultimately create family dysfunction contributing to suicidality.

In addition to some common universal factors like mental disorders, substance abuse disorder, family history of mental disorders, family history of suicide, chronic medical illness and problems of family relationship [33–36], this study added some other risk factors such as being member of a joint family, being ≥26 years of age at the first marriage, having twice or more marriage, having conjugal life of ≥16 years, having no child and ≥3 children. Unemployment, living in urban area, low level of education, being married, low socioeconomic status are also found to contribute to causation of suicidality in the Indian subcontinent countries [54–56].

Bangladesh is one of those countries where the surveillance system for suicide attempts is limited [6]. An estimated number of 20 suicide attempts result in one completed suicide [6] and as we have found in our study and many other studies, suicidal ideation is highly correlated to suicide attempt. These findings indicate that long-term surveillance and monitoring of suicidality and its risk factors can provide necessary information for the development of strategies for the prevention of suicidal deaths [6]. The combination of information on suicidality and suicidal deaths can help estimate case fatality rates that would ultimately assist in identifying high-vulnerable individuals [6,9]. Another important aspect is that there is a lack of internationally standardized protocol for data collection regarding suicide attempts that create methodological differences in the collection of data and surveillance [57]. A uniform approach is required for surveillance of both reported and unreported events of suicidality, and legal status and procedures of suicide and suicide attempt should also be considered [57].

## Effect of suicidal ideation on suicide attempt

This study substantiated the fact that suicidal ideation was highly correlated to suicide attempt showing higher odds of likelihood of committing SA in the future as was revealed in many other previous studies in different populations [15,16].

## Strength and limitations

Perhaps, this was the first time we attempted to study on suicidality among urban married adults in Bangladesh. Appropriate statistical models were used in this study that revealed some important findings. However, there are limitations too. The data was not nationally representative. Some variables like spousal violence methods of suicide attempts and self-perceived reason for SI and SA were not considered in this study. Since the data were self-reported, some events of SI and SA might be kept hidden that could produce underestimated results. The percentage of suicidality was very low, our selected logistic regression model was under-powered, we need more sample for getting achieve adequate power. Moreover, substance abuse was entirely self-reported and no validated scales were used, and the type of substance abused was unknown. More research is required on suicidality and suicide attempts among adults in Bangladesh.

## Conclusions

This study aimed to identify the risk factors of suicidality among married adults in Rajshahi City, Bangladesh. A total of 708 samples were recruited for this study. The prevalence of suicidal ideation, suicide attempt, and suicidality was 5.8%, 3.4%, and 8.3% respectively. Joint family, ≥26 years of age at the first marriage, twice or more marriage, conjugal life of ≥16 years, having no child and ≥3 children, medical comorbidity, mental comorbidity, stress-anxiety, family history of mental disorders, family history of suicide/attempt, substance abuse, poor relationship with the spouse, and poor relationship with other family members were the most influential predictors of suicidality. Moreover, it was observed that suicidal ideation was the most important risk factor for suicidal attempt. The concerned government and non-

government agencies can use the findings of this study to identify the vulnerable groups and individuals and undertake measures for preventing and reducing suicidality and suicide in the country.

## Supporting information

**S1 Data.**
(SAV)

## Acknowledgments

The authors would like to express their heartfelt thanks to all who participated in or otherwise contributed to this research.

## Author Contributions

**Conceptualization:** Abdul Wadood, Golam Hossain.

**Data curation:** Rezaul Karim, Masud Rana.

**Formal analysis:** Abdul Wadood, Masud Rana, Golam Hossain.

**Investigation:** Rezaul Karim, Golam Hossain.

**Methodology:** Abdul Wadood, Golam Hossain.

**Resources:** Abdullah Al Mamun Hussain.

**Software:** Golam Hossain.

**Supervision:** Golam Hossain.

**Validation:** Abdul Wadood, Rezaul Karim, Abdullah Al Mamun Hussain, Masud Rana, Golam Hossain.

**Visualization:** Masud Rana.

**Writing – original draft:** Abdul Wadood, Golam Hossain.

**Writing – review & editing:** Abdul Wadood, Rezaul Karim, Abdullah Al Mamun Hussain, Masud Rana, Golam Hossain.

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
