## [Decision Letter · Decision Letter 0]

22 Jan 2021

PONE-D-20-35857

Risk factors of suicidality among married adults: A cross-sectional survey in Rajshahi City, Bangladesh

PLOS ONE

Dear Dr. Hossain,

Thank you for submitting your manuscript to PLOS ONE. After careful consideration, we feel that it has merit but does not fully meet PLOS ONE’s publication criteria as it currently stands. Therefore, we invite you to submit a revised version of the manuscript that addresses the points raised during the review process.

We look forward to receiving your revised manuscript.

Kind regards,

Vincenzo De Luca

Academic Editor

PLOS ONE

Journal Requirements:

Reviewers' comments:

Reviewer's Responses to Questions

**Comments to the Author**

1. Is the manuscript technically sound, and do the data support the conclusions?

Reviewer #1: Yes

2. Has the statistical analysis been performed appropriately and rigorously? 

Reviewer #1: No

3. Have the authors made all data underlying the findings in their manuscript fully available?

Reviewer #1: No

4. Is the manuscript presented in an intelligible fashion and written in standard English?

Reviewer #1: Yes

5. Review Comments to the Author

Reviewer #1: Dear authors,

I consider the research topic important as suicide is a major public health concern in Bangladesh. I appreciate your work and I encourage you to address my comments and recommendations that could potentially increase the impact of your findings.

Major comments:

Introduction

• I recommend that the authors include more details about suicidality in Bangladesh i.e., sociocultural context, incidence, prevalence. In my opinion, the authors focused too much on US studies and relevant sociocultural differences exist between the US and Bangladesh. It would be useful for readers if the authors provided a synthesis of the main differences in suicidality between western countries and the Indian subcontinent.

• In my opinion, the introduction should contain a synthesis of the main factors that are known to influence suicidality with an emphasis on relevant factors in the Indian subcontinent.

• The authors provided the general aim of the study, but I recommend that they clearly state the objectives.

Methods

• I encourage the authors to provide more details related to the sample size calculation and how this translates into results interpretation. Since analyses included participants (two per household), how does the expected margin of error assist the reader in interpreting results i.e., suicidality, suicidal ideation and attempts? Why did the authors not calculate the sample size using the total population of Rajshahi? It is not clear from their sample calculation if the logistic models are adequately powered.

• I am unsure what was the rationale behind calculating a sum score for suicidality. The possible values for suicidality are 0, 1 or 2. Since you used binary logistic regression, please clarify what were the values of suicidality used in the calculation. I did not understand following statement because it is unclear how you calculated the percentile: “A percentile of ≥75% was considered as suicidality and <75% as no suicidality” (lines137-138).

• In my opinion, it would be important to describe how you measured substance abuse i.e., use of validated scales. Did the authors measure substance use or substance abuse?

• For exploring the correlates of suicide attempt, the authors could use a stepwise modelling approach (and enter in the step 1 sociodemographic variables and in step 2 suicidal ideation) and provide the variation attributable to suicidal ideation. As a suggestion, the authors could use “No” as reference category for suicidal ideation.

Results

Based on the percentage of suicidality of 8.3% your logistic regression model is under-powered as you need about 1600 observations to achieve adequate power. Peduzzi et al. recommended following formula to calculate the minimum sample for a binary logistic regression model n=10k/p where n=sample size; k=number of independent variables (14 in your model) and p the smallest proportion in the binary model (in this case 0.083). Therefore, more work is needed to find the most relevant variables for the final model e.g., based on societal characteristics), both for the analysis of suicidality and suicide attempts.

Discussion

• I recommend that the authors elaborate more on the differences in suicidality, suicide intentions and attempts found in their study compared to other studies conducted in western countries and the Indian subcontinent. In my opinion more work is needed to synthesize differences in relevant risk factors for suicidality compared to other studies.

Strengths and limitations,

• The authors may consider including in the limitations the fact that the study included a relatively small sample of participants, considering the low prevalence of suicidality/suicide attempts. Another limitation worth mentioning is that the authors did not include relevant risk factors of suicidality such as spousal violence, see Naved RT, Akhtar M “Spousal Violence Against Women and Suicidal Ideation in Bangladesh”, Women’s Health Issues 18 (2008) 442–452

Minor comments:

• Some statements lack clarity and need revising e.g., Introduction, lines 63-64 “A study found that SA recorded 0.79% of the lifetime prevalence and 0.29% of the past 12-month prevalence”

• Methods: For clarity, please revise the description of the number of questionnaires analysed. The authors mention on line 126: “Thus, a total of 139 samples were excluded from the study” and I am unsure if samples mean households or participants. If the authors excluded 139 households then data was collected from 311 households (i.e., 450-139=311) and 622 participants (311*2 as 2 adults per household were interviewed). Therefore, I am unsure why the authors mentioned that 761 individuals participated.

• Methods: Please provide the rationale for asking about suicidal ideation and attempts in the last 6 months instead of the last 12 months (frequently used in this field)

• Methods, Table 1-in my opinion this should be merged with results and contain percentages for each category (variable); providing the coding is less informative. I suggest that the authors provide explanations whether they measured gender identity or biological sex. For improving clarity, following variables should be better explained: “homemaker”; “hard-worker” (also called headworker, line160), “nuclear” and “joint” family. Please clarify if the variable “Respondent’s suicidal thought/attempt” is equivalent with suicidality.

• Results: I suggest that the authors describe the substances of use/abuse (including the main substance used if the information is available) and the medical and mental comorbidities.

• Results provided in table 2 and Table 3 could be merged to give the reader a better overview of the relevance of the variables in bivariate and multivariate analyses.

• Results, lines 179-193, since most results are provided in Table 3, the paragraph could be shortened

• Please provide the number of observations included in the binary logistic regression models

Relevant articles related to suicide in Bangladesh:

Jordans et al. BMC Psychiatry (2014) 14:358. “Suicide in South Asia: a scoping review” DOI 10.1186/s12888-014-0358-9

Saidur Rahman Mashreky , Fazlur Rahman & Aminur Rahman (2013) Suicide

Kills More Than 10,000 People Every Year in Bangladesh, Archives of Suicide Research, 17:4,

387-396, DOI: 10.1080/13811118.2013.801809

Murad M. Khan. Suicide on the Indian Subcontinent. Crisis 2002; Volume 23 (3): 104–107. DOI: 10.1027//0227-5910.23.3.104

6. PLOS authors have the option to publish the peer review history of their article (what does this mean?). If published, this will include your full peer review and any attached files.

Reviewer #1: No

---

## [Author Response · Author response to Decision Letter 0]

14 Mar 2021

Response to Reviewers 

Journal Name: PLOS ONE

Tracking No. ( Manuscript ID): PONE-D-20-35857

 Manuscript Title: “Risk factors of suicidality among married adults: A cross-sectional survey in Rajshahi City, Bangladesh"

Dear Editor

Thank you very much for providing reviewer’s comments on our manuscript. We have modified and revised the manuscript accordingly, and detailed corrections point–by-point is given below:

Review Comments to the Author: 

Reviewer #1: 

I consider the research topic important as suicide is a major public health concern in Bangladesh. I appreciate your work and I encourage you to address my comments and recommendations that could potentially increase the impact of your findings.

Authors’ Response: Thank you for your kind efforts for improving our manuscript. We appreciate your comments and suggestions on the manuscript. 

Major comments:

Introduction

• I recommend that the authors include more details about suicidality in Bangladesh i.e., sociocultural context, incidence, prevalence. In my opinion, the authors focused too much on US studies and relevant sociocultural differences exist between the US and Bangladesh. It would be useful for readers if the authors provided a synthesis of the main differences in suicidality between western countries and the Indian subcontinent. In my opinion, the introduction should contain a synthesis of the main factors that are known to influence suicidality with an emphasis on relevant factors in the Indian subcontinent.

Authors’ Response: According to your suggestions, we added some new information and facts about suicide in Indian subcontinent with Bangladesh [Page 5, Line 82-90; Page 6, Line 97-111]. 

• The authors provided the general aim of the study, but I recommend that they clearly state the objectives.

Authors’ Response: According to your recommendation, we added specific objectives [Page 7, Line 119-122.

Methods

• I encourage the authors to provide more details related to the sample size calculation and how this translates into results interpretation. Since analyses included participants (two per household), how does the expected margin of error assist the reader in interpreting results i.e., suicidality, suicidal ideation and attempts? Why did the authors not calculate the sample size using the total population of Rajshahi? It is not clear from their sample calculation if the logistic models are adequately powered.

Authors’ Response: You know, this was a household study, and we considered married couple who were living together in Rajshahi city as a study population. We got the information about total number of households in Rajshahi city from Rajshahi city corporation office. Since we had the total number of households, we used the mentionable formula to calculate the required number of households (sample, married couple living in household). We have described in details about the sample size determination in Page, 7; Line: 132-135. 

We think no need to mention statistical model which would be used for analyzing data like Logistic regression in subtitle “Sample size determination and Sampling” if we consider a formula for calculating sample size which is not related to a specific statistical model. 

• I am unsure what was the rationale behind calculating a sum score for suicidality. The possible values for suicidality are 0, 1 or 2. Since you used binary logistic regression, please clarify what were the values of suicidality used in the calculation. I did not understand following statement because it is unclear how you calculated the percentile: “A percentile of ≥75% was considered as suicidality and <75% as no suicidality” (lines137-138).

Authors’ Response: We measured suicidality by addition of suicidal Ideation and Suicide Attempt. That is why we calculated the sum score. You correctly assumed that values for suicidality are 0, 1 or 2. We could assume 0 for No Suicidality and 1-2 for Suicidality for using binary logistic regression. But we followed a new idea of making a cut-off point at 75% of sum score of Suicidality and a percentile of ≥75% of the sum score was considered as suicidality and <75% of the sum score as no suicidality”. 

• In my opinion, it would be important to describe how you measured substance abuse i.e., use of validated scales. Did the authors measure substance use or substance abuse?

Authors’ Response: We measured substance abuse by a question ‘Are/were you addicted?’ with two options of answer ‘yes’ and ‘no’. We did not use any scale for measuring substance abuse.

• For exploring the correlates of suicide attempt, the authors could use a stepwise modelling approach (and enter in the step 1 sociodemographic variables and in step 2 suicidal ideation) and provide the variation attributable to suicidal ideation. As a suggestion, the authors could use “No” as reference category for suicidal ideation.

Authors’ Response: Thank you for your suggestion. We considered “yes” as reference case for keeping similarity of binary logistic and stepwise logistic results.

Results

Based on the percentage of suicidality of 8.3% your logistic regression model is under-powered as you need about 1600 observations to achieve adequate power. Peduzzi et al. recommended following formula to calculate the minimum sample for a binary logistic regression model n=10k/p where n=sample size; k=number of independent variables (14 in your model) and p the smallest proportion in the binary model (in this case 0.083). Therefore, more work is needed to find the most relevant variables for the final model e.g., based on societal characteristics), both for the analysis of suicidality and suicide attempts.

Authors’ Response: We followed a publication where authors used the formula that we used in our present study for calculating sample size, and binary logistic regression was used in that paper [Please see, “Assessment of knowledge regarding tuberculosis among non-medical university students in Bangladesh: a cross-sectional study”] but we agree with you. Yes, more work is needed regarding this issue, we have mentioned in Limitation [Page, 20-21; Line: 333-336]. 

Discussion

• I recommend that the authors elaborate more on the differences in suicidality, suicide intentions and attempts found in their study compared to other studies conducted in western countries and the Indian subcontinent. In my opinion more work is needed to synthesize differences in relevant risk factors for suicidality compared to other studies.

Authors’ Response: We have tried to follow your suggestion and revised some parts of Discussion [Page 19, Line 303-309].

Strengths and limitations

• The authors may consider including in the limitations the fact that the study included a relatively small sample of participants, considering the low prevalence of suicidality/suicide attempts. Another limitation worth mentioning is that the authors did not include relevant risk factors of suicidality such as spousal violence, see Naved RT, Akhtar M “Spousal Violence Against Women and Suicidal Ideation in Bangladesh”, Women’s Health Issues 18 (2008) 442–452

Authors’ Response: We have added the suggested limitations [Page 20, Line 331-332].

Minor comments:

• Some statements lack clarity and need revising e.g., Introduction, lines 63-64 “A study found that SA recorded 0.79% of the lifetime prevalence and 0.29% of the past 12-month prevalence”

Authors’ Response: This sentence was revised [Page 4, Line 63-65].

• Methods: For clarity, please revise the description of the number of questionnaires analyzed. The authors mention on line 126: “Thus, a total of 139 samples were excluded from the study” and I am unsure if samples mean households or participants. If the authors excluded 139 households then data was collected from 311 households (i.e., 450-139=311) and 622 participants (311*2 as 2 adults per household were interviewed). Therefore, I am unsure why the authors mentioned that 761 individuals participated.

Authors’ Response: We selected 450 households and targeted to collect information from two married people (one male and one female) from one household, i.e., the number of respondents would be 900. But 73 disagreed to give interviews, and 66 were not available in their households at the time of the survey, thus, a total of 139 (73+66=139) respondents could not be interviewed. The rest 761 (900-139=761) respondents were interviewed. For the sake of clarity we revised this portion [Page 8, Line 156-160].

• Methods: Please provide the rationale for asking about suicidal ideation and attempts in the last 6 months instead of the last 12 months (frequently used in this field)

Authors’ Response: Actually this study is a part of a project entitled “Household study on dual burden of bipolar disorders among couples of Rajshahi city in Bangladesh”. For that project time of reference for all information was last 6 months. That is why we could not avoid 6 months instead of 12 months. 

• Methods, Table 1-in my opinion this should be merged with results and contain percentages for each category (variable); providing the coding is less informative. I suggest that the authors provide explanations whether they measured gender identity or biological sex. For improving clarity, following variables should be better explained: “homemaker”; “hard-worker” (also called headworker, line160), “nuclear” and “joint” family. Please clarify if the variable “Respondent’s suicidal thought/attempt” is equivalent with suicidality.

Authors’ Response: Table 1 has been revised according to your suggestion. Actually, we did not use the variable Respondent’s suicidal thought/attempt”, so we have deleted from the Table. 

• Results: I suggest that the authors describe the substances of use/abuse (including the main substance used if the information is available) and the medical and mental comorbidities.

Authors’ Response: These specific information were not available. So we could not mention or describe these.

• Results provided in table 2 and Table 3 could be merged to give the reader a better overview of the relevance of the variables in bivariate and multivariate analyses.

Authors’ Response: Thank you for your suggestion. You know, we put Chi-square test and logistic regression results in Table 2 and Table 3 respectively. In this study, Chi-square test was used to select the independent variables for logistic model. We think no need to merge Table 2 and Table 3. 

• Results, lines 179-193, since most results are provided in Table 3, the paragraph could be shortened. 

Authors’ Response: We have shortened the said paragraph [Page 14; Line: 214-225].

• Please provide the number of observations included in the binary logistic regression models

Authors’ Response: We provided the said number [Page 14, Line 211].

Relevant articles related to suicide in Bangladesh:

Jordans et al. BMC Psychiatry (2014) 14:358. “Suicide in South Asia: a scoping review” DOI 10.1186/s12888-014-0358-9

Mashreky SR, Rahman F, Rahman A. Suicide Kills More Than 10,000 People Every Year in Bangladesh, Archives of Suicide Research (2013); 17:4, 387-396, DOI: 10.1080/13811118118.2013.801809.

Murad M. Khan. Suicide on the Indian Subcontinent. Crisis 2002; Volume 23 (3): 104–107. DOI: 10.1027//0227-5910.23.3.104

Authors’ Response: Thank you for proving some articles which are related to our current study. We have studied your suggested articles and used necessary information in Introduction and Discussion of our manuscript. 

We would like to thank the reviewers for the valuable comments. We have revised the documents to the best of our ability, but we will definitely be happy to provide further improvement if there are further clarifications required. 

With best regards

Dr. Md. Golam Hossain

Professor of Health Research Group

Department of Statistics, University of Rajshahi

Rajshahi-6205, Bangladesh

E-mail: hossain95@yahoo.com

---

## [Decision Letter · Decision Letter 1]

6 Apr 2021

PONE-D-20-35857R1

Risk factors of suicidality among married adults: A cross-sectional survey in Rajshahi City, Bangladesh

PLOS ONE

Dear Dr. Hossain,

Thank you for submitting your manuscript to PLOS ONE. After careful consideration, we feel that it has merit but does not fully meet PLOS ONE’s publication criteria as it currently stands. Therefore, we invite you to submit a revised version of the manuscript that addresses the points raised during the review process.

We look forward to receiving your revised manuscript.

Kind regards,

Vincenzo De Luca

Academic Editor

PLOS ONE

Journal Requirements:

Reviewers' comments:

Reviewer's Responses to Questions

**Comments to the Author**

1. If the authors have adequately addressed your comments raised in a previous round of review and you feel that this manuscript is now acceptable for publication, you may indicate that here to bypass the “Comments to the Author” section, enter your conflict of interest statement in the “Confidential to Editor” section, and submit your "Accept" recommendation.

Reviewer #1: (No Response)

2. Is the manuscript technically sound, and do the data support the conclusions?

Reviewer #1: Yes

3. Has the statistical analysis been performed appropriately and rigorously? 

Reviewer #1: Yes

4. Have the authors made all data underlying the findings in their manuscript fully available?

Reviewer #1: Yes

5. Is the manuscript presented in an intelligible fashion and written in standard English?

Reviewer #1: No

6. Review Comments to the Author

Reviewer #1: Dear authors,

I thank the authors for considering my suggestions and for revising the manuscript accordingly. While most of the comments provided in the first round have been addressed, some issues merit further consideration:

1. Abstract

• As a suggestion, if the authors need to save space, they could only include AORs without CIs and p values as these are provided in the results section.

2. Introduction:

• In my opinion, the authors could further synthesize the information presented in the introduction (instead of enumerating the results of each study) to make the case for the necessity of their work and guide the reader towards the objectives.

• Objectives 2 and 3 are very similar and could be combined.

3. Methods:

• I regret but from the provided information I do not understand how you calculated the sum score. For each observation, possible values are 0, 1 or 2. Therefore, I am unsure what represents the 75% cutoff.

• The authors could acknowledge as a limitation that substance abuse was entirely self-reported and no validated scales were used. Also, the authors could mention as a limitation that the type of substance abused is unknown.

• In the statistical analysis section (contrarily to the response provided to my comments) the authors do not mention that they used stepwise regression.

• Please provide in table 3 the number of observations that was used to run the multivariate model. The number of completed interviews is 761 and it is important to specify if the model ran on 761 observations or some observations were excluded because of missing values

• Since the present study is part of a larger project, I suggest that the authors provide this information in the methods section

• The authors stated that they revised table 1 (according to my suggestions) but the coding column was not changed, and percentages are not provided.

7. PLOS authors have the option to publish the peer review history of their article (what does this mean?). If published, this will include your full peer review and any attached files.

Reviewer #1: No

---

## [Author Response · Author response to Decision Letter 1]

9 Apr 2021

Response to Reviewers 

Journal Name: PLOS ONE

Tracking No. ( Manuscript ID): PONE-D-20-35857R1

Manuscript Title: “Risk factors of suicidality among married adults: A cross-sectional survey in Rajshahi City, Bangladesh"

Dear Editor

Thank you very much for providing reviewer’s comments/suggestions on our manuscript. We have modified and revised the manuscript accordingly, and detailed corrections point–by-point is given below:

Review Comments to the Author: 

Reviewer #1: 

I thank the authors for considering my suggestions and for revising the manuscript accordingly. While most of the comments provided in the first round have been addressed, some issues merit further consideration:

Authors’ Response: Thank you for your kind efforts for improving our manuscript. We appreciate your comments and suggestions on the manuscript. 

1. Abstract

• As a suggestion, if the authors need to save space, they could only include AORs without CIs and p values as these are provided in the results section.

Authors’ Response: Thank you for your suggestion. We have deleted 95% CIs, and considered AORs and p-values, p-values have been considered for justification of significant. 

2. Introduction:

• In my opinion, the authors could further synthesize the information presented in the introduction (instead of enumerating the results of each study) to make the case for the necessity of their work and guide the reader towards the objectives.

Authors’ Response: According to your advice, we revised some portions of the introduction section [Line 55-92].

• Objectives 2 and 3 are very similar and could be combined.

Authors’ Response: We have combined objectives 2 and 3 as 2 [Line: 119-120].

3. Methods:

• I regret but from the provided information I do not understand how you calculated the sum score. For each observation, possible values are 0, 1 or 2. Therefore, I am unsure what represents the 75% cutoff.

Authors’ Response: Authors’ Response: In our present study, suicidality was measured by addition of two items (i) suicidal ideation (yes=1; no=0) and (ii) suicide attempt (yes=1; no=0). We calculated the sum of score (the sum of score would be 0, or 1, or 2 for a particular respondent). We classified our sample into two groups; (i) No suicidality (sum of score 0; code 0), and (ii) Suicidality (Sum of Score 1-2; code 1). We have described this issue in Line: 167-170. 

For another issue (cut-off point 75%), actually, we followed a publication where authors used a cut-off point at 75% of sum score of depression’ and a percentile of ≥75% of the sum score was considered as depression and <75% of the sum score as no depression”. 

We have reanalyzed our data considering the categorical outcome variable, suicidality (Yes=1, No=0), and got same logistic regression results as we considered 75% cut-off points. We have deleted 75% cut-off point from text to avoid confusion [Line: 170-172]. 

• The authors could acknowledge as a limitation that substance abuse was entirely self-reported and no validated scales were used. Also, the authors could mention as a limitation that the type of substance abused is unknown.

Authors’ Response: We agree with you, and have put these issues in limitation [Line: 339-341]. 

• In the statistical analysis section (contrarily to the response provided to my comments) the authors do not mention that they used stepwise regression.

Authors’ Response: Actually, stepwise logistic regression model was not used in our present study, so we did not need to mention it in Statistical Analysis. We used multivariable (multiple) logistic regression model for analyzing our data, and mentioned it in Statistical Analysis[Line:185-186]. . 

• Please provide in table 3 the number of observations that was used to run the multivariate model. The number of completed interviews is 761 and it is important to specify if the model ran on 761 observations or some observations were excluded because of missing values. 

Authors’ Response: We have mentioned in Method [Line: 160-161], finally 708 samples were used in our present study. Since, we did not have missing values of any variable, we have mentioned (n=708) in the title of Table 3 and Table 2. 

• Since the present study is part of a larger project, I suggest that the authors provide this information in the methods section. 

Authors’ Response: We have mentioned this issue in Method [Line: 129-130]. 

• The authors stated that they revised table 1 (according to my suggestions) but the coding column was not changed, and percentages are not provided.

Authors’ Response: Earlier, you suggested, “Table 1-in my opinion this should be merged with results and contain percentages for each category (variable); providing the coding is less informative”. According to your earlier suggestions, we have revised Table 1 accordingly, and put it in Result section. Table 2 also has been revised. 

We would like to thank the reviewer for the valuable comments. We have revised the documents to the best of our ability, but we will definitely be happy to provide further improvement if there are further clarifications required. 

With best regards

Dr. Md. Golam Hossain

Professor of Health Research Group

Department of Statistics, University of Rajshahi

Rajshahi-6205, Bangladesh

E-mail: hossain95@yahoo.com

---

## [Decision Letter · Decision Letter 2]

3 May 2021

Risk factors of suicidality among married adults: A cross-sectional survey in Rajshahi City, Bangladesh

PONE-D-20-35857R2

Dear Dr. Hossain,

We’re pleased to inform you that your manuscript has been judged scientifically suitable for publication and will be formally accepted for publication once it meets all outstanding technical requirements.

Kind regards,

Vincenzo De Luca

Academic Editor

PLOS ONE

Additional Editor Comments (optional):

Reviewers' comments:

Reviewer's Responses to Questions

**Comments to the Author**

1. If the authors have adequately addressed your comments raised in a previous round of review and you feel that this manuscript is now acceptable for publication, you may indicate that here to bypass the “Comments to the Author” section, enter your conflict of interest statement in the “Confidential to Editor” section, and submit your "Accept" recommendation.

Reviewer #1: All comments have been addressed

2. Is the manuscript technically sound, and do the data support the conclusions?

Reviewer #1: Yes

3. Has the statistical analysis been performed appropriately and rigorously? 

Reviewer #1: Yes

4. Have the authors made all data underlying the findings in their manuscript fully available?

Reviewer #1: Yes

5. Is the manuscript presented in an intelligible fashion and written in standard English?

Reviewer #1: Yes

6. Review Comments to the Author

Reviewer #1: I consider your manuscript a valuable addition to the existing literature. All the best in your future research.

7. PLOS authors have the option to publish the peer review history of their article (what does this mean?). If published, this will include your full peer review and any attached files.

Reviewer #1: No

---

## [Editor Report · Acceptance letter]

4 May 2021

PONE-D-20-35857R2 

Risk factors of suicidality among married adults: A cross-sectional survey in Rajshahi City, Bangladesh 

Dear Dr. Hossain:

I'm pleased to inform you that your manuscript has been deemed suitable for publication in PLOS ONE. Congratulations! Your manuscript is now with our production department. 

Kind regards, 

on behalf of

Dr. Vincenzo De Luca 

Academic Editor

PLOS ONE